# mHealth Interventions to Promote a Healthy Diet and Physical Activity among Cancer Survivors: A Systematic Review of Randomized Controlled Trials

**DOI:** 10.3390/cancers14153816

**Published:** 2022-08-06

**Authors:** Lufan Wang, Crystal S. Langlais, Stacey A. Kenfield, June M. Chan, Rebecca E. Graff, Isabel E. Allen, Chloe E. Atreya, Erin L. Van Blarigan

**Affiliations:** 1Department of Epidemiology & Biostatistics, University of California, San Francisco, CA 94158, USA; 2Department of Urology, University of California, San Francisco, CA 94158, USA; 3UCSF Helen Diller Family Comprehensive Cancer Center, San Francisco, CA 94158, USA; 4Department of Medicine, University of California, San Francisco, CA 94158, USA

**Keywords:** digital health, behavioral interventions, nutrition, physical activity, cancer survivorship

## Abstract

**Simple Summary:**

The number of cancer survivors has increased dramatically in the past several decades. Research indicates that health behaviors, including having a healthy diet and engaging in regular exercise, may improve the treatment outcomes and quality of life of cancer survivors. Behavioral interventions using web and mobile technology may be feasible and acceptable approaches to modify physical activity and dietary behaviors. This review summarizes the feasibility, acceptability and estimated effects of physical activity and dietary interventions using web and mobile technology from the published studies.

**Abstract:**

*Background*: Technology-based interventions are increasingly used to improve physical activity (PA) and diet. *Methods*: We conducted a systematic review of randomized controlled trials (RCTs) published up to August 2021 that tested mobile health (mHealth) PA and/or dietary interventions among cancer survivors and reported on the feasibility, satisfaction, behavioral change, and/or quality of life (QOL) outcomes. *Results*: In total, 61 articles were identified on PubMed, and 23 of those met the inclusion criteria. The most common cancers were breast (*n* = 1000), prostate (*n* = 713), and colorectal (*n* = 650). Participants were predominantly White (median: 84%, interquartile range (IQR): 20%) and college-educated (58%). The interventions varied, but the most common combination of components (six studies) was a website/mobile app with an activity tracker and coaching. In terms of duration, 70% (*n* = 16) of the interventions lasted 12 weeks. The median total tracker wear was 87% of the study days (IQR: 6%) and the median text-message reply rate was 73% (IQR 4%). Most participants (median: 87%; IQR: 16%) were satisfied with at least one intervention component. Eleven out of 18 studies examining behavioral change reported significant between-group differences and six out of 11 studies examining QoL reported significant improvements. *Conclusions*: mHealth interventions are a promising approach to improving the PA and diets of cancer survivors. Research in racially/ethnically and socioeconomically diverse populations is needed.

## 1. Introduction

As of January 2022, it was estimated that there were 18 million cancer survivors in the United States (US), and the prevalence of cancer in the US is projected to approach 26 million by 2040 [1]. Early detection and improvement in treatments have led to improved survival rates for cancer. Cancer survivors often deal with physical effects of cancer and its treatments, such as fatigue and pain, as well as psychological effects of cancer, ranging from fear of recurrence to anxiety and depression. Research indicates that health behaviors, including a healthy diet and regular physical activity, are associated with better treatment outcomes, quality of life (QoL), and overall survival in people with cancer [2,3,4,5,6]. In 2022, the American Cancer Society (ACS) updated its nutrition and physical activity guidelines for cancer survivors. These guidelines include being physically active and maintaining a healthy eating pattern that is high in vegetables, fruits, and whole grains and low in red and processed meats, sugar-sweetened beverages, highly processed foods, and refined grain products [7]. Similarly, the World Cancer Research Fund/American Institute for Cancer Research updated their expert report on diet, nutrition, physical activity, and cancer in 2018. In this report, the emphasis of their recommendations shifted toward an integrated pattern of behaviors [8]. Despite the known benefits of a healthy diet and physical activity, adherence to these recommendations is low among cancer survivors [9].

Mobile health (mHealth) is defined by the World Health Organization (WHO) as ‘medical and public health practices supported by a mobile device, such as mobile phone, patient monitoring devices, personal digital assistants and other wireless devices’ [10]. mHealth is becoming increasingly common in healthcare and represents a promising approach for increasing physical activity and modifying dietary behaviors. mHealth tools include, but are not limited to, websites, email, mobile applications (apps), text messaging, and wearable activity trackers. These tools can support goal setting, self-monitoring, and instruction, as well as provide feedback on physical activity and diet change [11]. A growing body of research has incorporated mHealth into lifestyle interventions to increase physical activity and modify dietary behaviors among different populations [12,13,14]. The purpose of this article is to summarize published studies reporting on mHealth physical activity and/or dietary interventions among cancer survivors.

## 2. Methods

This systematic review is registered in Open Science (Registration DOI: 10.17605/OSF.IO/8EYC2) and was performed in accordance with the Preferred Reporting Items for Systematic Review and Meta-Analysis (PRISMA) guidelines [15].

In this review, we summarize findings from recent (in the past ~5 years) randomized controlled trials (RCTs) with mHealth interventions focused on healthy diet and/or physical activity promotion in cancer survivors, defined as any person who has been diagnosed with cancer [16]. We define an mHealth intervention as one that includes a website/mobile app, text messages, and/or activity trackers. Given how quickly mHealth technology changes, we used 2016 as the earliest cut-off date for our search. As shown in Figure 1, we used the following search terms to identify relevant titles and abstracts in PubMed: mHealth intervention (digital OR website OR text message OR app OR Fitbit OR “wearable device” OR tracker), cancer patients or cancer survivors, dietary and/or physical activity (lifestyle OR behavioral OR physical activity OR exercise OR diet OR nutrition), and intervention. We selected “clinical trial” as the article type in the PubMed search filters. A single author (LW) reviewed titles and abstracts of papers published from 1 January 2016 to 13 August 2021 and written in English that examined an mHealth intervention among adult cancer survivors (aged 18 or over) to determine eligibility (*n* = 61).

To be included, studies were required to meet the following criteria: a randomized control trial with adult participants (aged 18 or over) testing an mHealth physical activity and/or dietary intervention. Studies were excluded if *only* real-time telephone calls and/or video calls and/or non-automated text messaging were used as the intervention. One of the advantages of a mHealth intervention is reduced person-time required per patient (from staff or other interventionists). Therefore, we included an inclusion criterion that some aspect was not executed in real-time by a human. If the intervention included only direct coaching by phone, the intervention did not meet our definition of a digital health intervention and was excluded.

The results of the search and reasons for ineligibility can be found in Figure 1. Information from the titles and abstracts was used to determine whether the papers met our eligibility criteria. The most common reason for exclusion was an intervention that was not targeted at physical activity and/or dietary behavioral change or was not an mHealth intervention (*n* = 17). We also excluded studies with participants who were not diagnosed with cancer (*n* = 3), studies that were not randomized controlled trials (*n* = 4), and articles that included participants who were less than 18 years old (*n* = 3). Lastly, to focus on scalable mHealth interventions, we excluded two articles reporting interventions that included *only* real-time telephone calls, video calls, and/or non-automated text messaging (e.g., direct text messaging with a human). Of the remaining 32 articles that were determined eligible for a full-text review, 24 articles (22 studies, including two studies with different portions of their results each reported in two separate articles) were deemed eligible for this review (Furthermore, one article that was suggested by the reviewer but was not found by our search criteria, yet did meet the inclusion criteria, was added. The total number of studies included in this review, therefore, was 23).

Predefined data-extraction tables were used to summarize the study design and participant characteristics (Table 1), intervention characteristics and findings of behavioral change (Table 2), and findings concerning the feasibility of and satisfaction with the intervention (Table 3). A narrative approach was used to synthesize a study of the characteristics and key findings of the included evidence [17].

The quality of the study design was assessed by a trained reviewer (LW) for each study using a scoring system adapted from a review of eHealth interventions [18,19,20]. A score was assigned to each study based on the following nine methodological characteristics: individual randomization, use of a control group for comparison, testing a single technology, use of pre-/posttest design, participant retention, equivalence of baseline groups, handling missing data, sample size calculation, and validity of measures. The range of possible scores was 0–100%. Studies were not excluded based on their quality scores.

**Table 1 cancers-14-03816-t001:** Randomized controlled trials of mHealth interventions focused on a healthy diet and physical activity among cancer survivors (2016–2021).

Author, Year	Country	Sample Size	Study Population	Duration of Intervention	Intervention	Comparison Group	Primary Outcomes	Other Outcomes	Follow-Up Schedule: Overall Completion
Rees-Punia et al., 2021 [21]	US	85	Stage I or II breast, colon, endometrium, kidney, or bladder cancer survivors who reported less than 150 min MVPA/week and/or less than 2 days/week of strength training	12 weeks	Intervention (n = 45): access to the HEALED website including physical activity training recommendations, exercise videos, and a goal-setting tool; received monthly email reminders to return to the website	Waitlist control (n = 40)	Feasibility, acceptability, and usability	Physical activity and sedentary time	0 and 12 weeks: 92%
Pinto et al., 2021 [22]	US	20	Stage I–III >65-year-old breast (n = 15) or other (n = 5) cancer survivors who had completed treatment within 5 years	12 weeks	Intervention (n = 12): tailored step goal program with recommendation of listening to audiobooks during physical activity	Control (n = 8): tailored step goal program without audiobook recommendation	Difference in daily steps	Light-intensity physical activity, sedentary time, MVPA; intervention evaluation	0 and 12 weeks: 95%
Chan et al., 2020 [23]	US	202	Prostate cancer survivors	12 weeks	Level 2 (n = 51): level 1 plus personalized diet and exercise prescription delivered through website; Level 3 (n = 50): level 2 plus Fitbit Alta with physical activity reports, text messages (4 per week), and weekly survey for progress tracking; Level 4 (n = 52): level 3 plus 2 optional 30-min calls, one with an exercise trainer and one with a dietitian	Level 1 (n = 49): information about exercise and diet, resource directory, and guidelines delivered by website	Feasibility and acceptability; satisfaction; frequency of intervention use	Self-reported physical activity and diet; lifestyle behavior score	0, 3, and 6 months: 83% at 3 months and 77% at 6 months
Chow et al., 2020 [24]	US	41	Adults more than 5 years from initial cancer diagnosis who received hematopoietic cell transplantation or with any history of acute leukemia or lymphoma	16 weeks	Intervention (n = 24): 30-min telephone-based review session; Fitbit tracker and healthwatch360 app; coaching on goal-setting and feedback on their activity or diet by email or text message; private Facebook peer support group; iCanQuit app for smokers	Control (n = 17): 30-min telephone-based review session; Fitbit tracker and healthwatch360 app	Feasibility	Physical activity; HRQoL	0 and 16 weeks: 90%
Nguyen et al., 2020 [25]	Australia	83	Inactive, post-menopausal, stage I–III breast cancer survivors	12 weeks	Intervention (n = 43): wrist-worn Garmin vivofit2, behavioral feedback and goal setting in a single face-to-face session, and five telephone-delivered behavioral counseling sessions	Waitlist control (n = 40): abridged intervention with activity tracker at 12 weeks	Sleep quality		0, 12, and 24 weeks: 94% at 12 weeks and 86% at 24 weeks
Rastogi et al., 2020 [26] ^a^ Cadmus-Bertram et al., 2019 [27] ^a^	US	50 dyads	Stage I–III breast (n = 45) or colorectal (n = 5) cancer survivors who had finished primary treatment and the survivors’ support partners	12 weeks	Intervention (n = 25 dyads): Fitbit tracker, educational handbook; survivors and partners were asked to assist each other; coaching email sent by staff at weeks 1, 2, 4, and 8	Control (n = 25 dyads): 2015 US Dietary Guidelines for Americans; standardized emails at 1, 2, 4, and 8 weeks with information on healthy eating and stress management	QoL and sleep; physical activity	Intervention feedback	0 and 12 weeks: 94%
Van Blarigan et al., 2020 [28]	US	50	Stage I–IV colorectal cancer survivors who were not actively undergoing chemotherapy and were considered disease-free or had a stable disease status	12 weeks	Intervention (n = 25): printed materials and personalized diet report; orientation session to website; website with dietary goal setting, food tracking, summary, progress, recipes, and meal planning; text messages (one per day)	Waitlist control (n = 25): print materials from weeks 1–12; had option to receive intervention from weeks 12–24 after completing 12-week assessment	Feasibility and acceptability	Self-reported diet; technician-assessed body measures	0, 3, and 6 months: 90% at 3 months and 84% at 6 months
Finlay et al., 2020 [29]	Australia	71	Stage I–III prostate cancer survivors who had completed primary treatment	4 weeks	Two intervention arms received the same computer-tailored physical activity self-monitoring and feedback modules; arms differed in the website architecture; Standard tunneled arm (n = 27) received a single weekly module that combined ‘once-off’ advice with a physical activity log; Free-choice arm (n = 27) received the ‘once-off’ tailored advice modules as standalone modules that could be accessed at any time and in any order.	Control (n = 17): access to homepage of website with static information	Differences in completion rates of the four physical activity logs between the two intervention arms	Website usage, physical activity, and user perceptions	0 and 4 weeks: 70%
Gell et al., 2019 [30]	US	66	Stage I–III breast (n = 38) or other cancer (n = 28) survivors who had completed a supervised oncology rehabilitation program with no concurrent radiation or chemotherapy	8 weeks	Intervention (n = 34): Fitbit tracker, health coach session about physical activity at week 1, follow-up calls from health coach at weeks 2, 4, 8, and 25; 25 text messages over 8 weeks	Control (n = 32): Fitbit tracker only	Physical activity measured by accelerometer	Participants’ satisfaction with the Fitbit, health coach session, and text messages	0 and 8 weeks: 89%
Mohamad et al., 2019 [31]	UK	62	Overweight or obese prostate cancer survivors who were not currently enrolled in a weight management program and had no distant metastases	12 weeks	Intervention (n = 31): one group meeting, a supporting letter from their urologist, three telephone dietitian consultations at 4-week intervals; a pedometer; access to web-based diet and physical activity resources	Waitlist control (n = 31): delayed intervention group session and option to receive Fitbit and access to website after 12 weeks	Difference between groups in change in body weight at 12 weeks and 12 months	HRQoL; feasibility and acceptability	0, 3, 6, and 12 months: 87% at 3 months, 82% at 6 months, and 44% at 12 months
Maxwell-Smith et al., 2019 [32]	Australia	68	Stage I or II colorectal (n = 53) or endometrial (n = 15) cancer survivors at cardiovascular disease risk who had completed active cancer treatment within 5 years	12 weeks	Intervention (n = 34): Fitbit tracker; two group sessions about physical activity with a behavioral change specialist at weeks 1 and 4; 20-min phone call at week 8 for support and feedback	Control (n = 34): printed materials on physical activity guidelines	Minutes per week of MVPA; cardiovascular risk markers (blood pressure, BMI)		0 and 12 weeks: 94%
Dong et al., 2019 [33]	China	60	Stage I –III post-operative breast cancer survivors with no metastasis, mental health problem, or dementia who had finished adjuvant radiotherapy/chemotherapy within 4 months to 2 years	12 weeks	Intervention (n = 30): face-to-face televideo muscle training (three/week) and endurance training (four/week); mobile app to record step counts; daily prompt with information on postoperative breast cancer rehab and physical exercise rehab from social media app	Control (n = 30): traditional treatment and rehab	HRQoL	Muscle strength; cardiorespiratory capacity	0 and 12 weeks: 83%
Gomersall et al., 2019 [34]	Australia	36	Colorectal (n = 25), prostate (n = 10) or breast (n = 1) cancer survivors with no cardiopulmonary or metabolic disorders at least 1 month post-surgery	Standard clinic, 4 weeks; text message-enhanced clinic, 12 weeks	Text message-enhanced clinic (n = 18): standard clinic plus tailored text messages for 12 weeks. Frequency of text messages varied with a minimum of six per fortnight. The purpose of the messages was to reduce sedentary time and increase MVPA	Face-to-face standard clinic (n = 18): participated in four 1-h clinical exercise rehab program with AEP sessions; home exercise information	Feasibility and acceptability; daily time spent sitting	Sitting, standing, stepping at a light or moderate-to-vigorous intensity, sedentary behavior	0 and 12 weeks: 86%
Kenfield et al., 2019 [35]	US	76	Stage T1–T3a nonmetastatic prostate cancer survivors who had completed treatment more than 3 months before enrollment	12 weeks	Intervention (n = 37): Fitbit, personalized recommendation report based on eight healthy behaviors; access to website and Fitbit community group; one email every 2 weeks and four to five text messages /week on four areas: get active, eat well, stop smoking, find support	Standard of care control (n = 39)	Feasibility and acceptability	Change in the prostate score of 8 and individual behaviors; objective change in MVPA and daily steps; body size; HRQoL; maintenance or adoption of behaviors after 1 year	0, 3, and 12 months: 84% at 3 months and 64% at 12 months
McNeil et al., 2019 [36] ^b^ McNeil et al., 2021 [37] ^b^	Canada	45	Stage I–IIIc breast cancer survivors who had completed adjuvant treatment, except for hormonal therapy	12 weeks	Instruction of lower-intensity physical activity (n = 15) or higher-intensity physical activity (n = 15); completed diary with responses to questions and goal-setting every 3 weeks; active follow-up discussion by phone or email with exercise physiologist	Control (n = 15): instruction to maintain baseline physical activity levels	Total physical activity, MVPA and light-intensity physical activity, and sedentary and sleeping times	Health-related fitness (body size, body scan, fitness); adherence to the prescribed physical activity interventions; total absolute physical activity time; ≥40% heart rate reserve	0, 3, and 6 months: 96% at 3 months and 91% at 6 months
Van Blarigan et al., 2019 [38]	US	42	Stage I–III colorectal cancer survivors, who were disease-free at enrollment	12 weeks	Intervention (n = 21): printed materials on physical activity after cancer, Fitbit flex with website providing physical activity feedback, daily automated text messages (one per day)	Control (n = 21): printed educational materials about physical activity after cancer	Feasibility and acceptability	Changes in moderate physical activity, MVPA, vigorous physical activity, and daily steps between baseline and 12 weeks	0 and 12 weeks: 93%
Ferrante et al., 2018 [39]	US	35	Stage 0–III breast cancer survivors who identified as African American and had a BMI ≥ 25	6 months	Intervention (n = 18): had access to SparkPeople website; received handout with goal information on weight loss, calorie intake, and physical activity; a wrist-worn physical activity tracker (Fitbit); 30-min training on using the website	Waitlist control group (n = 17) received handout with goal information on weight loss, calorie intake, and physical activity; a wrist-worn physical activity tracker (Fitbit)	Adherence and acceptability	Weight change; QoL	0, 6, and 12 months: 97% at 6 months and 89% at 12 months
Ormel et al., 2018 [40]	Netherlands	32	Testicular (n = 27), breast (n = 4) or osteosarcoma (n = 1) cancer survivors	12 weeks	Intervention (n = 16): Information about benefits of regular physical activity; instructed to self-monitor physical activity with RunKeeper and activate training reminder in the app	Usual care control (n = 16)	Change in physical activity between baseline, 6 weeks, and 12 weeks	App usability and patients’ experience	0, 6, and 12 weeks: 100% at 6 and 12 weeks
Golsteijn et al., 2018 [41]	Netherlands	478	Prostate (n = 292) or colorectal (n = 186) cancer survivors receiving adjuvant treatment (at least 6 months post-surgery) or who had successfully completed primary treatment up to 1 year ago	12 weeks	Intervention (n = 249): Computer-tailored physical activity advice at three time points and pedometer; access to interactive content on the website	Waitlist control group (n = 229)	Change in physical activity	HRQoL; fatigue; distress	0, 3, and 6 months: 89% at 3 months and 87% at 6 months
Mayer et al., 2017 [42]	US	284	Post-cancer treatment, inactive stage I–III colon cancer survivors	6 months	Intervention (n = 144): received all materials provided to the controls; smartphones with the SurvivrosCHESS application that included core skill building, support services, and information services and tools; a coach was available in the later study period to initiate a discussion group and send tailored private message to inactive users	Control (n = 140): received National Cancer Institute’s “Facing Forward: Life after Cancer Treatment” Booklet, the National Coalition for Cancer Survivorship’s Cancer Survival Toolbox, and a pedometer	Change in MVPA at 6 months	Distress; QoL	0 and 6 months: 80% at 6 months
Valle et al., 2017 [43]	US	35	Stage I–IIIA African American/Black breast cancer survivors with a BMI of 20–45 who had completed cancer treatment	6 months	Intervention group (n = 13): individual face-to-face session with information about weight; received a Bluetooth- and WIFI-enabled wireless scale that connected to a mobile app and website and were instructed to weigh themselves daily; received 24 weekly emails that delivered behavioral lessons and tailored feedback on their weight; Intervention+ group (n = 11): received the intervention above plus an activity tracker and tailored feedback on their physical activity, a website/app, and a behavior lesson twice per week without tailored feedback	Control group (n = 11) had an initial group session and received a wireless scale with a companion app	Proportion of participants who completed the 3- and 6-month assessments	Anthropometric and clinical measures (weight, waist circumference, body composition, etc.); adherence to self-monitoring; adherence to weight-management strategies; diet and physical activity; acceptability and satisfaction	0, 3, and 6 months: 94% at 3 months and 97% at 6 months
Short et al., 2017 [44]	Australia	492	Stage I–IV breast cancer survivors who had finished active cancer treatment and were not already meeting national physical activity guidelines	12 weeks	Group A (n = 167): three tailored modules with information and interactive feedback on physical activity, one per month; Group B (n = 168): three modules with information and interactive feedback on physical activity in the first three weeks of the 12-week intervention period, one per week; Group C (n = 157): single module with information on physical activity but no interactive feedback in the first week of the 12-week intervention period; All participants had access to an action planning tool and information on resistance training		Engagement with website	Website acceptability and self-reported physical activity	0, 3, and 6 months: 32% at 3 months and 11% at 6 months
Gnagnarella et al., 2016 [45]	Italy	125	Breast (n = 77), gastrointestinal (n = 20), gynecologic (n = 8), lung (n = 6), or other (n = 14) cancer patients not receiving enteral nutrition, parental nutrition or palliative care and not reporting significant weight loss in the last 6 months	6 months	Intervention (n = 61): access to an interactive nutritional online information website with social media features	Control (n = 64): PDF version of the website content by email	Change in nutritional knowledge	HRQoL and psychological distress inventory	0 and 24 weeks: 54%

Abbreviations: BMI, body mass index; MVPA, moderate-to-vigorous physical activity; HRQoL, health-related quality of life; QoL, quality of life; RD, registered dietitian; AEP, accredited exercise physiologist. ^a^ [26,27] described the same study. ^b^ [36,37] described the same study.

**Table 2 cancers-14-03816-t002:** Combination of mHealth intervention tools used in randomized controlled trials among cancer survivors to promote physical activity and/or dietary change, and associations with change in physical activity, diet, and/or quality of life, sorted by outcome measure.

Author, Year	Sample Size	Website/Mobile App	Wearable Activity Tracker	Coaching	Text Message	Result	
						Between-Group Difference	Within-Group Difference
**Lifestyle score based on diet and** **physical activity**							
Chan et al., 2020 [23]	202	√	√	√	√	Significant between-group differences in mean **lifestyle score** change compared to level 1 were observed in levels 2 (mean change: 0.9, 95% CI: 0.4–1.4), 3 (mean change: 0.5, 95% CI: 0.02–1.0), and 4 (mean change: 1.1, 95% CI: 0.7–1.6) at 12 weeks	Not reported
Kenfield et al., 2019 [35]	76	√	√		√	Improvements in four out of eight **recommended behaviors** were observed in the intervention arm. The estimated mean **lifestyle score** of the intervention arm was 1.5 (95% CI: 0.7–2.3) points higher than that of the control arm at 12 weeks (*p* < 0.001)	Median (IQR) absolute changes in the lifestyle score from the baseline to 12 weeks were 2 (1, 3) points in the intervention arm and 0 (1, 1) points in the control arm
**Physical activity**							
Chow et al., 2020 ^a^ [24]	41	√	√	√	√	No significant between-group difference in physical activity over time was observed	No significant within-group change was observed
Pinto et al., 2021 [22]	20	√	√	√		Significant differences in the changes in **step counts** (Cohen’s effect size = 1.1, *p* = 0.02) and **MVPA** (Cohen’s effect size = 1.0, *p* = 0.04) favoring the audiobook group were observed from 0 to 12 weeks	Participants in the audiobook group on average added 1487.2 **steps per day** (Cohen’s effect size = 0.79, *p* = 0.11) and 71 min per week of **MVPA** (Cohen’s effect size = 0.8, *p* = 0.15) from 0 to 12 weeks
Cadmus-Bertram et al., 2019 ^b^ [27]	50 dyads	√	√	√		Compared to the control group, survivors in the intervention group had a significant improvement in the **MVPA** minutes per week (effect size = 1.1, *p* < 0.01) and daily steps (effect size = 1.0, *p* < 0.01) at 12 weeks	Survivors in the intervention group increased their MVPA by 69 ± 84 min/week and daily steps by 1470 ± 1881. Survivors in the control group decreased their MVPA by 20 ± 71 min/week and daily steps by 398 ± 1751
Maxwell-Smith et al., 2019 [32]	68	√	√	√		Improvement in minutes of **MVPA** per week (mean difference in change: 66 min/week, *p* = 0.03) in the intervention group compared to the control group at 12 weeks	Intervention group increased their MVPA by 45 min/week (95% CI: 2–88), while a reduction of 21 min/week (95% CI: −59–17) was observed for the control group
Mayer et al., 2017 [42]	284	√	√	√		No significant between-group difference in physical activity over time was observed	Not reported
Valle et al., 2017 [43] ^a^	35	√	√	√		No differences between groups over time by way of a change in dietary intake or energy expenditure from physical activity at 3 months	A significant increase in **energy expenditure** from the baseline to 6 months was observed in the intervention group with the wearable activity tracker (median: 432, IQR: 706, *p* = 0.03)
Van Blarigan et al., 2019 [38]	42	√	√		√	No difference in the change in physical activity was found from the baseline to 12 weeks between arms	Not reported
Ferrante et al., 2018 [39] ^a^	35	√	√		√	No between-group difference in physical activity was observed	No within-group change was observed
Golsteijn et al., 2018 [41]	478	√	√			Participants in the intervention group improved their self-reported **MVPA** minutes per week (between-group change: 139, 95% CI: 9.4–269.0, *p* = 0.04) and days with at least 30 min of physical activity in a week (between-group change: 0.8, 95% CI: 0.5–1.1, *p* < 0.01) at 3 months, and ActiGraph assessed **MVPA** (between-group change: 45.9, 95% CI: 13.5–78.3, *p* < 0.01) at 6 months	Not reported
Rees-Punia et al., 2021 [21]	85	√				No between-group difference in sedentary, light-intensity physical activity, MVPA, or self-reported strength training	Not reported
Ormel et al., 2018 [40]	32	√				Significant median difference in change in self-reported physical activity score favoring the intervention group (median: 12.1, IQR: 105.1, *p* = 0.02) at 6 weeks	Not reported
Short et al., 2017 [44]	492	√				No between-group difference in physical activity	Significant improvements in **self-reported MVPA** minutes were observed in the single-module group (mean change: 192.3, 95% CI: 139.9–244.8), weekly module group (mean change: 168.3, 95% CI: 116.8–219.9), and monthly module group (mean change: 173.9, 95% CI: 119.0–228.8)
Finlay et al., 2020 [29]	71	√				No between-group differences in self-reported MVPA or resistance training	An increase across groups in the percentage of participants meeting the guidelines relative to the baseline scores (free choice +25%; tunneled +20%; control +36%). Within-group changes in MVPA in all groups were not statistically significant
Gell et al., 2019 [30]	66		√	√	√	Difference in change in weekly **MVPA** minutes between groups (*p* = 0.03; effect size d = 0.6) observed at 8 weeks	Intervention group maintained their weekly MVPA minutes (mean change: 26.2, *p* = 0.35) while the control group had a significant decrease in their weekly MVPA minutes (mean change: −57.5, *p* = 0.03)
McNeil et al., 2019 [36]	45		√	√		Increase in **MVPA** (between-group difference: 36, 95% CI: 6–60, *p* < 0.01) min/day and decrease in **sedentary** (between-group difference: −72, 95% CI: −132 to −12, *p* = 0.02) min/day were significantly greater in the lower-intensity physical activity group compared to the control group at 12 weeks. No significant differences were noted between the high-intensity physical activity and control groups	Adjusted mean increases in total (mean change: 42, 95% CI: 6–78, *p* = 0.02) min/day and **MVPA** (mean change: 24, 95% CI: 6–42, *p* = 0.01) min/day were observed in the high-intensity physical activity group
Gomersall et al., 2019 [34]	36			√	√	Compared to the standard group, the text message-enhanced group showed a significant improvement in self-reported **MVPA** at 4 weeks (between-group difference: 53.2 min/day, 95% CI: 2.9–103.5, *p* = 0.04). By 12 weeks, relative to the standard group, participants in the text message-enhanced group sat less (between-group difference: −80.1 min/day, 95% CI: −156 to −3.8, *p* = 0.04) and participated in more MVPA (between-group difference: 67.3 min/day, 95% CI: 24.0–110.6, *p* = 0.02)	Compared to the baseline, participants in the text message-enhanced group engaged in more vigorous physical activity (mean change: 19.6 min/day, 95% CI: 2.5–36.8), and participants in the standard clinic engaged in less MVPA (mean change: −50 min/day, 95% CI: −79.1 to −21.1, *p* < 0.01)
**Diet**							
Chow et al., 2020 [24] ^a^	41	√	√	√	√	No significant between-group difference over time observed	No significant within-group change observed
Van Blarigan et al., 2020 [28]	50	√			√	Compared to the control arm, the intervention arm had a significant improvement **in whole grain consumption** at 12 weeks (between-group difference: 0.9 servings/d, 95% CI: 0.1–1.6)	Not reported
Ferrante et al., 2018 [39] ^a^	35	√	√		√	No between-group difference observed	No within-group change observed
Valle et al., 2017 [43] ^a^	35	√	√	√		No differences between groups over time in changes in dietary intake	No within-group difference over time in dietary intake in any study group
Gnagnarella et al., 2016 [45]	125	√				No between-group difference observed	Nutritional questionnaire score improved in both groups
**Quality of life (QoL)**							
Chow et al., 2020 [24]	41	√	√	√	√	No between-group difference in QoL over time observed	Compared to the baseline, significant improvements in physical (mean change: 2.7, 95% CI: 0.7–4.6) and mental health (mean change: 4.2, 95% CI: 1.5–6.9) were observed in the intervention group at 16 weeks
Rastogi et al., 2020 [26] ^b^	50 dyads	√	√	√		Relative to the control group, the intervention group was associated with a moderate-to-large improvement in **physical** (effect size: 0.4, 95% CI 0.0–0.8) and **mental health** (effect size: 0.6, 95% CI 0.2–1.0) at 12 weeks	Compared to the baseline, significant improvements in aggregate physical health scores (mean change: 4.3, 95% CI: 0.2–8.4), mental health (mean change: 4.0, 95% CI: 1.5–6.5), role limitation due to emotional problems (mean change: 3.7, 95% CI: 0.1–7.2), and vitality (mean change: 6.1, 95% CI: 3.3–8.9)
Mohamad et al., 2019 [31]	62	√	√	√		Significant difference in **overall QoL score** changes between the intervention and control groups observed at 12 weeks (between-group difference: 12.3, 95% CI 4.9–19.7, *p* < 0.01)	Compared to the baseline, a significant improvement in functioning subscales was observed in the intervention group (mean change: 4.0, 95% CI 0.4–7.5, *p* = 0.03) while a significant decrease in the overall QoL score (mean change: −5.1, 95% CI −10.1 to −0.1) was observed in the mini-intervention (control) group at 12 weeks
Mayer et al., 2017 [42]	284	√	√	√		No between-group difference in QoL over time was observed	Not reported
Golsteijn et al., 2018 [41]	478	√	√			Compared to the control group, a significant decrease in **fatigue** (between-group difference: −3.7, 95% CI: −6.8 to −0.5, *p* = 0.02) and improvement in **physical functioning** (between-group difference: 2.3, 95% CI: 0.5–4.1, *p* = 0.01) were observed in the intervention group	Not reported
Kenfield et al., 2019[35]	76	√	√		√	No difference in change in QoL measures between intervention and control groups	Not reported
Ferrante et al., 2018 [39]	35	√	√		√	No between-group difference in physical activity observed	Significant improvement in **QoL** observed in the intervention group (mean change: −9.4, 95% CI: −10.4 to −7.6, *p* = 0.03)
Dong et al., 2019 [33]	60	√		√		Differences in change from the baseline favoring the intervention group in terms of **vitality** (*p* < 0.01), **mental health** (*p* < 0.01), and **reported health transition** (*p* < 0.01) when comparing the two groups at 12 weeks	Within-group change in role—physical (mean change: 25.0, *p* < 0.01), vitality (mean change: 5.2, *p* = 0.01), and mental health (mean change: 3.5, *p* = 0.01)—of SF-36 observed in intervention group from baseline to 12 weeks
Gnagnarella et al., 2016 [45]	125	√				Difference in **role functioning score** change observed (*p* = 0.02)	Increased overall QoL scores observed in both groups
McNeil et al., 2019 [36]	45		√	√		No difference in self-reported QoL noted across time or between groups in this study	Not reported
Nguyen et al., 2020 [25]						Greater reductions in actigraphy-based awake time (min) after sleep onset (mean difference: −5.7, 95% CI: −11.7 to −0.2) and number of awakenings (mean difference: −2.0, 95% CI: −3.6–0.4) in the intervention arm compared to the waitlist arm at 12 weeks	In the intervention group, significant improvements in waking (min) after sleep onset (mean change: 2.1, SD: 3.4, *p* < 0.01), number of awakenings (mean change: −1.0, SD: 1.2, *p* = 0.06), and total PSQI score (mean change: −0.8, SD: 0.4, *p* = 0.03) were observed at 12 weeks and sleep efficiency (mean change: 2.1, SD: 3.4, *p* < 0.01) at 24 weeks

Abbreviations: 95% CI, confidence interval; MVPA, moderate-to-vigorous physical activity; IQR, interquartile range; PSQI, Pittsburgh Sleep Quality Index; QoL, quality of life; SD, standard deviation. ^a^ These studies reported results for both physical activity and diet, separately. ^b^ [26,27] described the same study.

**Table 3 cancers-14-03816-t003:** Adherence to and acceptance of mHealth behavioral interventions tested in randomized controlled trials among cancer survivors.

Author, Year	Intervention Duration	Adherence to Intervention Components			Satisfaction/Acceptability
		Website/Mobile Application Usage	Wearable Activity Tracker Wearing Time	Text Messages Response Rate	
Finlay et al., 2020 [29]	4 weeks	Mean number of physical activity logs completed: 2.6 (SD:3) for tunneled arm, 1.5 (SD: 1.4) for free-choice arm	N/A ^a^	N/A ^a^	The self-reported engagement and relevance scores were low to moderate across groups, with no significant between-group differences
Gell et al., 2019 [30]	8 weeks	N/A ^a^	Participants wore the Fitbit an average of 6 or more days per week (≥86%) throughout the 8-week intervention	N/A Text messages did not ask for reply	In total, 91% of participants were satisfied or very satisfied with the Fitbit, while 93% and 90% of participants in the intervention group were satisfied with the health coaching component and the content of the text messages, respectively. The results showed that 91% of participants in the intervention group perceived the Fitbit as often or almost always motivating for physical activity, while 55% and 70% of participants in the intervention group reported text messages and the health coaching sessions as motivating to be physically active, respectively
Rees-Punia et al., 2021 [21]	12 weeks	Median number of logins per person over 12-week intervention: 4 (IQR: 7); median total time logged in: 95 min (IQR: 193)	N/A ^a^	N/A ^a^	Mean score for system usability scale: 72 (range: 67–78); mean ratings for motivation and enjoyment of the website: 3.8/5 (range: 1–4) and 3.6/5 (range: 2–5), respectively
Pinto et al., 2021 [22]	12 weeks	N/A ^a^	89% (n = 17) of participants wore Fitbit on >90% of the 84 study days	N/A ^a^	Overall, 89% (n = 16) of the participants were very satisfied with their participation; 100% (n = 19) of the participants found the Fitbit to be helpful for physical activity
Chan et al., 2020 [23]	12 weeks	Median number of days visiting the website for levels 1, 2, 3, and 4: 2 (IQR: 2), 9 (IQR: 8), 11 (IQR: 8), and 16 (IQR: 9), respectively, of 84 study days	Not reported	N/A Text messages did not allow for a response	Most were satisfied or very satisfied with the intervention: 51% (n = 20), 64% (n = 27), 52% (n = 23), and 64% (n = 27) for levels 1, 2, 3, and 4, respectively
Nguyen et al., 2020 [25]	12 weeks	N/A ^b^	N/A ^b^	N/A ^b^	N/A ^b^
Rastogi et al., 2020 * [26] Cadmus-Bertram et al., 2019 * [27]	12 weeks	44% of participants reported that they logged into the Fitbit website more than once per day, 13% logged in daily, 26% logged in 4–6 times per week, and 9% logged once per week or less	Not reported	N/A ^a^	Overall, 74% (n = 18) of the survivors in the intervention group were “extremely satisfied with the intervention”; 91% (n = 22) and 62% (n = 15) of the survivors in the intervention group rated Fitbit and coaching emails, respectively, as “very important” or “extremely important” in helping them to increase their physical activity
Van Blarigan et al., 2020 [28]	12 weeks	Median number of days participants visited the website: 13 (IQR: 32) out of 84 days	N/A ^a^	The intervention arm responded to a median 15 (IQR: 11) of 21 text messages that asked for a reply	In total, 74% (n = 31) of the participants were satisfied or very satisfied with the text messages; 64% (n = 28) of the participants were satisfied or very satisfied with the overall intervention
Mohamad et al., 2019 [31]	12 weeks	Median number of visits to the website: 5 (IQR: 12) for the intervention group, and 8 (IQR: 12) for the waitlist mini-intervention	Not reported	N/A ^a^	Of the samples, 58% (n = 15) of the participants in the intervention group and 46% (n = 13) of the participants in the control group accessed the online resource during the 12-week intervention
Maxwell-Smith et al., 2019 [32]	12 weeks	N/A ^b^	Average valid wear days ^c^ of Fitbit: 86% (SD:29) of 84 study days	N/A ^a^	N/A ^b^
Dong et al., 2019 [33]	12 weeks	N/A ^b^	N/A ^b^	N/A ^b^	N/A ^b^
Gomersall et al., 2019 [34]	12 weeks	N/A ^a^	N/A ^a^	The average reply rate to the fortnightly MVPA goal checks was 78% (n = 14) among the 18 participants in the intervention group	The average satisfaction score for the intervention, among the 17 participants in the intervention group who completed the satisfaction survey, was 4.3 (SD: 0.8) out of 5
Kenfield et al., 2019 [35]	12 weeks	Participants visited the website on a median of 3 days (IQR: 3) over the 12-week period	The intervention arm participants wore their Fitbit for a median of 82 (98% of the study days, IQR: 11) days in the 12-week period	The intervention arm responded to a median of 71% (IQR: 32%) of the 60 text messages that asked for a reply	Overall, 61% of the participants in the intervention arm rated the quality of the website as high or very high, 87% rated the Fitbits as good to excellent, and 69% rated the text messaging as good to excellent. Satisfaction (“satisfied” or “very satisfied”) for participants in the intervention arm was 60% for the website, 91% for Fitbits, and 73% for text messaging
McNeil et al., 2019 [36] McNeil et al., 2021 [37]	12 weeks	N/A ^a^	Not reported	N/A ^a^	Participants in the lower-intensity physical activity group enjoyed the intervention more than participants in the higher-intensity PA group (*p* = 0.05)
Van Blarigan et al., 2019 [38]	12 weeks	N/A ^a^	Participants in the intervention arm wore their Fitbits a median of 74 out of 84 days (88% of the study days, IQR 60)	Intervention arm participants responded to a median of 34 out of the 46 (74%; IQR: 25) text messages that asked for a reply	Overall, 88% (n = 14) of participants in the intervention arm were satisfied or very satisfied with their experience with the text messages and the Fitbit, reported that the text messages motivated them to exercise, and said that they would continue to wear the Fitbit after the study ended
Golsteijn et al., 2018 [41]	12 weeks	N/A ^b^	N/A ^b^	N/A ^a^	N/A ^b^
Ormel et al., 2018 [40]	12 weeks	Not reported	N/A ^a^	N/A ^a^	Among the 14 intervention participants who completed the qualitative semi-structured interview, 12 were enthusiastic about the RunKeeper app; 8 reported that they became more active due to the RunKeeper app and were planning to continue use of the app
Short et al., 2017 [44]	12 weeks	The average time spent on the study website was 57 min (SD: 72, range: 0–556) over the 12-week intervention The average number of visits to the website was 4.8 times (SD: 8.5, range: 1–146) over the 12-week intervention	N/A ^a^	N/A ^a^	Website acceptability among study completers was fair, with a mean score of 22.2 (SD: 5.98) out of a possible 36
Chow et al., 2020 [24]	16 weeks	92% of intervention participants interacted with the study’s mHealth apps	75% of the intervention participants met the goal ^d^ for regular fitness tracker use	N/A Text messages did not ask for reply	Among the 11 approached, 10 intervention participants expressed satisfaction with their experience
Ferrante et al., 2018 [39]	6 months	Mean number of days logged onto the website per week: 2.7 (95% CI: 2.2, 3.2)	Adherence with Fitbit was high; participants in both groups wore the Fitbit an average of 5 or more days per week (84%) throughout the 6-month intervention	N/A ^a^ Text messages did not ask for reply	Mean score for usefulness of the Fitbit: 4/4 (95% CI: 3.9–4.0)
Mayer et al., 2017 [42]	6 months	Among the participants in the intervention group, the median number of application uses was 15.7 (range: 1–27) throughout the possible 24 weeks of use	Not reported	N/A ^a^	N/A ^b^
Valle et al., 2017 [43]	6 months	Not reported	Among participants in the intervention group with a wearable activity tracker, the median total wear days was 162 (96.4%) out of the 168 study days	N/A ^a^	For participants in the intervention groups with and without a wearable activity tracker, respectively, the median acceptability scores for the smart scale were 4 (IQR: 1) and 2.5 (IQR: 2) out of 4; for the email feedback, they were 3 (IQR: 1.2) and 3 (IQR: 1) out of 4. For participants in the intervention group with a wearable activity tracker, the median acceptability score for the activity tracker was 4 (IQR: 1) out of 4
Gnagnarella et al., 2016 [45]	6 months	N/A ^b^	N/A ^a^	N/A ^a^	N/A ^b^

Abbreviations: IQR, interquartile range; SD, standard deviation; 95% CI, 95% confidence interval; MVPA, moderate-to-vigorous physical activity. * [26,27] are two articles describing the same study. ^a^ Tool was not used in the study. ^b^ Feasibility or acceptability is not the main outcome of interest in this study. ^c^ A step count of ≥1000 steps per day was defined as a valid wear-day. ^d^ ≥75% of the study days that the fitness tracker record ≥ 500 steps.

## 3. Results

### 3.1. Demographic Characteristics

Table 1 summarizes each study included in this review. Across the 23 unique studies, 2538 participants were enrolled, 54% of whom were female (*n* = 1359). Most studies (*n* = 16 described in 18 articles) included breast cancer, prostate cancer, and/or colorectal cancer survivors [23,25,26,27,28,29,31,33,34,35,36,37,38,39,41,42,43,44]. Besides these 16 studies, four studies included survivors of breast cancer and other cancer types (gynecologic cancer, testicular cancer, gastrointestinal cancer, lung cancer, osteosarcoma, and other rare cancers) [22,30,40,45], one included survivors of colorectal cancer and gynecologic cancer [32], one study included survivors of breast cancer, colorectal cancer, and other cancer types (endometrial cancer, bladder cancer, kidney cancer) [21], and one study included survivors of leukemia and lymphoma [24]. The most prevalent type of cancer diagnosis was breast cancer (*n* = 1000), followed by prostate cancer (*n* = 713) and colorectal cancer (*n* = 650). There were 175 participants diagnosed with other cancers (gynecologic cancer, testicular cancer, gastrointestinal cancer, leukemia, lymphoma, bladder cancer, kidney cancer, lung cancer, osteosarcoma, and others). Participants in 18 of the 23 studies had completed their primary cancer treatment before enrollment. 

The sample sizes of the 23 studies varied from 20 to 492 participants. Eighteen out of the 23 studies had a sample size less than 100. The mean age of participants across studies was 59.1 years (reported in 20 studies); the lowest and highest within-study mean age were 33.6 [40] and 71.6 years [22], respectively. Twelve of the 23 studies reported race/ethnicity [22,23,27,28,30,32,35,36,38,39,42,43]. In these 12 studies, participants were predominantly White [median: 84%, interquartile range (IQR): 20%]. Among the 15 studies [22,23,28,29,30,32,35,36,38,39,41,42,43,44,45] that reported information on the education of their participants, more than half (58%) of the participants were college-educated. Among the 10 studies [21,23,28,29,30,38,39,42,43,44] that reported participants’ work status, 49% of participants worked full- or part-time. 

### 3.2. Intervention Details

Of the 23 unique studies, 15 focused on physical activity only, two studies focused on diet only, and six studies targeted both physical activity and dietary change. Table 2 shows that, across the 23 interventions, 19 used websites/mobile apps, 15 included wearable activity trackers, 13 included in-person/telephone/video call/email coaching by study staff, and eight sent automated short message service (SMS) text messages to their participants. The most common combination of tools was a web/mobile app intervention with a wearable activity tracker and coaching by group session, email, or phone video call (*n* = 6) [22,26,27,31,32,42,43], followed by five studies that examined a web/mobile app intervention alone (*n* = 5) [21,29,40,44,45]. Across the eight studies that used text messages as one of the intervention tools, the frequency of text messages varied from once weekly to once daily. Four studies included text messages that asked for a reply.

Most studies used usual care and/or information concerning a healthy diet and/or physical activity as their comparator (control) intervention (*n* = 14). Six studies had waitlist controls where participants in the control group had the option to receive a delayed intervention [21,25,28,31,39,41]. Four studies [22,24,30,39] had Fitbit-only controls. The duration of the intervention ranged from 4 weeks to 6 months; the most common duration was 12 weeks (*n* = 16). Retention in the studies ranged from 32% to 100%, with a mean retention of 86%. The lowest retention, 32%, was reported by Short et al. from a study that tested a 12-week web-only intervention, aiming to examine different delivery schedules of physical activity advice modules among breast cancer survivors [44]. Studies with an intervention that included a wearable activity tracker had the highest mean retention rate (91%), and studies with an intervention that included a website/app only had the lowest (72%).

### 3.3. Feasibility and Acceptability

Feasibility (adherence) was reported in 19 out of the 23 studies (Table 3). It was defined differently across studies, including as website/mobile app usage, the wear time of wearable activity trackers, and/or response rates to text messages. Among the 19 studies that included a website/mobile app, six studies [21,23,28,31,44] with 12-week interventions reported the median number of days participants visited the website/mobile app; these numbers varied from 2% to 15% of the 84 study days. One study by Finlay et al. measured website usage by the number of physical activity logs completed across one control and two intervention arms [29]. A higher number of completed weekly physical activity logs was observed in the intervention arm (mean: 2.6 ± 1.3) that received an intervention content module weekly for four weeks than in the intervention arm (mean: 1.5 ± 1.4) that had access to all the intervention modules at any time and in any order. Adherence to wearable activity trackers was reported in eight studies [22,24,30,32,35,38,39,43] out of 15 that included wearable activity trackers. Adherence levels were generally high, with a median 87% (IQR: 6%) of study days wearing the devices. All four studies that included interactive text messages reported the text message response rate [28,34,35,38]. Overall, participants were responsive, with the median reply level being to 73% (IQR: 4%) of the text messages that asked for a reply.

Acceptability was measured in 17 out of the 23 studies (Table 3). It was mainly measured by semi-structured interviews or surveys to assess the participants’ perceptions of, and satisfaction with, the overall intervention, wearable activity trackers, text messages, and/or website/mobile app. Among the 12 studies that measured satisfaction, most participants were satisfied or very satisfied with at least one of the intervention components (median 87%, IQR: 16%). In five studies, participants perceived wearable activity trackers to be helpful and important for physical activity [22,27,30,39,43]. Four [28,30,35,38] out of the eight studies that sent regular text messages to participants assessed the participants’ satisfaction with the text messages, and 73% to 90% of the participants in these four studies were satisfied with the text messages and/or agreed that text messages motivated them to be physically active. In the studies by Ormel et al. [40] and Van Blarigan et al. [38], more than three-quarters of the participants in the intervention group (79% and 88%, respectively) said they would continue to use the mobile app or wear the Fitbit after the study ended. In Valle et al.’s study, all 24 participants in the intervention group would recommend the program with an in-person individual coaching session providing information on weight gain and the use of a wireless scale, along with the wireless scale itself, 24 weekly email-delivered behavioral lessons, and an optional activity tracker [43]. 

### 3.4. Behavioral Change

A change in physical activity and/or dietary behaviors was estimated in 18 of the 23 studies. As shown in Table 2, 16 studies estimated the change in physical activity [21,22,24,27,29,30,32,34,36,38,39,40,41,42,43,44], five studies reported the change in the diet [24,28,39,43,45], and two studies reported the change in a lifestyle score based on both physical activity and diet [23,35].

Among the 16 studies that reported physical activity change [21,22,24,27,29,30,32,34,36,38,39,40,41,42,43,44], six reported an intervention arm having demonstrated an increase in step counts [22,27] and physical activity time, ranging from 45 to 345 min per week [22,27,32,34,36,40]. Two studies reported that the intervention arm maintained a weekly MVPA from pre- to post-intervention [30,41]. These improvements were significantly different from the control arms. The intervention components in the eight studies with significant between-group differences in physical activity varied. Three studies included a web/mobile app with a wearable activity tracker and coaching [22,27,32]. Four studies used at least two of these tools in their intervention (wearable activity tracker + coaching + text messages [30]/website + wearable activity tracker [41]/wearable activity tracker + coaching [36]/coaching + text message [34]). One study tested an intervention with a mobile app only [40].

Five studies reported a dietary change [24,28,39,43,45]. One testing a web-based dietary intervention with daily text messages among colorectal cancer survivors found significantly greater improvement in whole grain consumption, measured using dietary records collected with the National Cancer Institute’s Automated Self-Administered Dietary Assessment Tool (ASA-24) [46] in the intervention arm at the end of the 12-week intervention compared to the control arm [28]. This improvement was maintained at a 24-week follow-up. The other four studies observed no change in diet measured by a self-administered nutrition questionnaire that was developed by Gnagnarella et al. [45]. food frequency questionnaires [47] at clinic visits, automated self-administered 24-hour dietary recalls (ASA-24) [46], or 24-h diet recalls administered by the research assistant using the Sparkpeople.com food diary tool [39]. Of those four, one study tested a web-based intervention with online nutrition information among cancer patients who were not receiving enteral nutrition, parenteral nutrition, or palliative care, and not reporting significant weight loss in the last 6 months [45]. One tested an mHealth intervention with a 30-min telephone session, a wearable activity tracker, a dietary tracker, and coaching on goal-setting and feedback by email or text messages [24]. The other two were weight-management programs among African American breast cancer survivors. One tested a program with an individual coaching session, a wireless scale, 24 weekly email-delivered behavioral lessons, and an optional activity tracker [43]. The other estimated the effect of a program with an interactive website, a wearable activity tracker, and text messages [39].

Significant improvement was reported by both studies that measured physical activity and dietary change using a composite lifestyle score [23,25]. These two studies targeted both physical activity and dietary change among prostate cancer survivors. One of the studies used a web-based intervention with a wearable activity tracker, text messages, and optional telephone coaching [23]. The other used a web-based intervention with a wearable activity tracker and text messages [35].

### 3.5. Quality of Life

Among the 11 studies that examined changes in QoL, six studies observed improvements (Table 2) [25,26,31,33,41,45]. One study with a web-based intervention with wearable activity trackers and coaching that targeted both diet and physical activity found significantly greater improvement in the overall QoL score measured by the European Organization for Research and Treatment of Cancer’s Quality of Life Questionnaire-C30 (EORTC QLQ-C30) in the intervention arm at the end of the 12-week intervention compared to the control arm [31]. Another study with a web-based intervention, a wearable activity tracker, and coaching on physical activity reported significantly greater improvement in physical and mental health measured by the SF-36 Health Survey [26]. Greater mental health improvement was also observed in one study using a mobile app and coaching in an intervention focused on physical activity, measured by the SF-36 Health Survey [33]. A study with a web-based physical activity intervention and wearable activity trackers reported greater improvement in both physical functioning measured by the EORTC QLQ-C30 and fatigue measured by Checklist for Individual Strength (CIS) [41]. Greater role functioning improvement, measured by the EORTC QLQ-C30, was observed in one study with a dietary intervention that included text messages and coaching [45]. Lastly, improvements in sleep quality, including less waking after sleep onset and a reduced number of awakenings, were observed in one study with a wearable activity tracker, one face-to-face session, and five telephone-delivered behavioral counseling sessions [25].

## 4. Assessment of Risk of Bias

The study design quality score for each study is presented in Table 4. The mean study design quality score was 75%. All included studies were randomized studies with control groups. Behavior/QoL outcomes were assessed both pre- and post-intervention, and validated measurement of outcomes was used in all studies. Retention was above 80% for all but three studies [29,44,45]. Thirteen out of 23 studies reported a sample size calculation. Fourteen out of 23 studies conducted the data analysis with consideration of the impact of the missing data. Eight out of 23 studies conducted a test to confirm the balance of baseline characteristics between the study groups. However, only seven out of 23 studies were designed to test the effectiveness of an isolated piece of technology [21,22,29,34,39,40,45]; all other studies were designed to test the effectiveness of a combination of intervention tools.

## 5. Discussion

The results of the 23 studies (25 publications) reviewed here provide evidence of the feasibility and acceptability of using mHealth interventions to promote behavioral change (diet and/or physical activity) among cancer survivors. Among the 23 studies, most focused on physical activity (*n* = 15) or targeted both physical activity and the diet (*n* = 6), while only two studies focused on the diet alone. More studies with interventions focused specifically on the diet are needed to assess the feasibility and acceptability, and improve the effectiveness, of mHealth dietary interventions. Additionally, only four out of the 23 studies evaluated 6-month interventions; the duration of the rest of the studies ranged from 4–16 weeks. Thus, the feasibility, adherence, and acceptability of these interventions over a longer period are unknown. 

Text messaging was commonly used as part of mHealth interventions, in combination with other components. Most of the text messages focused on providing tailored health promotion information and behavioral prompts. Personalized text messages with dietary behavior or physical activity information and reminders can motivate and support a change of behavior. Text messages that solicit a reply may increase participants’ engagement [20]. However, there is a lack of consensus or conclusive evidence from this review regarding the optimal frequency and timing of text messages. 

Wearable activity trackers, alone or in combination with other mHealth tools, were a feasible method to increase physical activity. Wearable activity trackers provide objective measures of physical activity and exercise [48]. They can also prompt behavioral change in real-time, assist users to self-monitor their physical activity, and provide automated feedback and rewards. These are behavioral change techniques associated with positive physical activity changes [49]. Adherence to wearing activity trackers was high, suggesting these devices are feasible and acceptable to participants. However, there was no standardized method for reporting wearable activity tracker outcomes, including adherence, validity, and physical activity measures [50]. The heterogeneous reporting of methods and results among studies using wearable physical activity trackers makes it difficult to compare findings across studies.

All included studies evaluated short-term (6 months or less) effects of mHealth interventions in relatively small sample sizes. The longer-term effects of mHealth interventions on maintaining physical activity and/or dietary behavioral change are unknown [51]. Tools such as websites/mobile apps, text messages, and wearable activity trackers in mHealth interventions may be useful for providing ongoing monitoring and support to cancer survivors, but studies with longer intervention and follow-up periods are needed to assess whether participants maintain engagement with mHealth interventions over time.

Of the 23 included studies, only one focused on the older population (≥65 years) [22]. This group carries a severe and disproportionate burden of cancer since two-thirds of cancer survivors are aged 65 or older in the US [1]. Additionally, in this review, most of the participants identified as White across the 12 studies that reported race/ethnicity, and more than half of the participants had at least a college/university degree across the 15 studies that reported education information. The lack of racial/ethnic and socioeconomic diversity in published studies is a limitation. mHealth interventions hold promise for improving health among underserved populations through low-cost approaches since they can be largely automated and disseminate information effectively. However, access and technology literacy are potential barriers. Data from the Pew Research Center showed that, as of 2020, 85% of Americans own a smartphone [52]. While overall smartphone ownership is high, it varies based on age, household income, and educational attainment. Bommakanti et al. [53] reported that patients who were older, male, less educated, and/or had a lower annual income were less likely to own smartphones, and thus could miss out on mHealth interventions requiring personal smartphone ownership. Patients might also be unwilling or unable to engage with mHealth interventions due to low smartphone literacy. On the other hand, a review from Armaou et al. [54] supported the effectiveness of web-based interventions to improve health in racial/ethnic minority and historically underserved communities. Studies have shown that linguistic and cultural tailoring can improve the effectiveness of health promotion interventions in minority or underserved populations [55,56]. Overall, more research is needed to assess the feasibility and acceptability of mHealth interventions in underrepresented populations. These interventions need to be tailored to the language and sociocultural characteristics of the target population.

Our review was limited in that all studies were identified through one database (PubMed). We also restricted our literature search to articles written in English. Therefore, relevant studies published in other databases or languages may have been missed. We did not exclude studies based on their quality scores. However, all included studies met at least five of the nine criteria. Additionally, as with other systematic reviews of published literature, there is the possibility of publication bias. In particular, four studies [23,28,30,38] listed QoL as one of their secondary or exploratory endpoints on clinicaltrial.gov, but had not yet reported results in the peer-reviewed literature at the time of our search. Of them, two studies reported their results on QoL in separate papers [57,58] published after our search date. 

## 6. Conclusions

Our results show that mHealth interventions are a promising approach to improving physical activity and dietary behaviors in cancer survivors. To better establish the optimal types and combination of mHealth interventions for cancer survivors, alternative study designs, as described by the Multiphase Optimization Strategy framework, may be useful [59]. Additionally, studies with larger sample sizes, longer study periods, and more racially/ethnically and socioeconomically diverse study populations are needed.

## Figures and Tables

**Figure 1 cancers-14-03816-f001:**
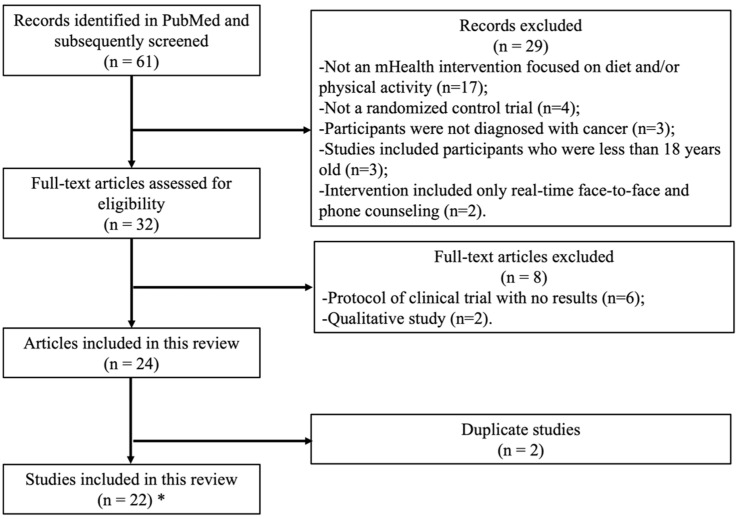
Flow chart of study selection. Search terms: (“2016/01/01”[Date—Publication]: “2021/08/13/”[Date—Publication]) AND (digital [tiab] OR website [tiab] OR text message [tiab] OR app [tiab] OR Fitbit [tiab] OR “wearable device”[tiab] OR tracker [tiab]) AND (lifestyle [tiab] OR behavioral [tiab] OR physical activity [tiab] OR exercise [tiab] OR diet [tiab] OR nutrition) AND (intervention [tiab]) AND (cancer [tiab]) AND (survivors OR patients). * One article that was suggested by the reviewer but was not found by our search criteria, though it did meet the inclusion criteria, was added. The total number of studies included in this review was 23.

**Table 4 cancers-14-03816-t004:** Summary of risk of bias among randomized controlled trials testing mHealth behavioral interventions in cancer survivors.

Author, Year	Individual Randomization	Control Group	Isolated Technology	Pre-/Posttest Design	Retention ≥80%	Baseline Equivalent Groups	Missing Data	Sample Size Calculation	Validated Measures	Score (% of Maximum)
Rees-Punia et al., 2021 [21]	Y	Y	Y	Y	Y	Y	N	NA	Y	78
Pinto et al., 2020 [22]	Y	Y	Y	Y	Y	Y	N	NA	Y	78
Chan et al., 2020 [23]	Y	Y	N	Y	Y	Unknown	Y	NA	Y	67
Chow et al., 2020 [24]	Y	Y	N	Y	Y	Unknown	Y	NA	Y	67
Nguyen et al., 2020 [25]	Y	Y	N	Y	Y	Unknown	Y	N	Y	67
Rastogi et al., 2020 * [26] Cadmus-Bertram et al., 2019 * [27]	Y	Y	N	Y	Y	Y	N	Y	Y	78
Van Blarigan et al., 2020 [28]	Y	Y	N	Y	Y	Unknown	N	Y	Y	67
Finlay et al., 2020 [29]	Y	Y	Y	Y	N	Unknown	Y	Y	Y	78
Gell et al., 2019 [30]	Y	Y	N	Y	Y	Y	Y	Y	Y	89
Mohamad et al., 2019 [31]	Y	Y	N	Y	Y	Unknown	Y	Y	Y	78
Maxwell-Smith et al., 2019 [32]	Y	Y	N	Y	Y	N	Y	Y	Y	78
Dong et al., 2019 [33]	Y	Y	N	Y	Y	Y	N	Y	Y	78
Gomersall et al., 2019 [34]	Y	Y	Y	Y	Y	Unknown	Y	Y	Y	89
Kenfield et al., 2019 [35]	Y	Y	N	Y	Y	Y	Y	Y	Y	89
McNeil et al., 2019 [36] 2021 [37]	Y	Y	N	Y	Y	Unknown	Y	Y	Y	78
Van Blarigan et al., 2019 [38]	Y	Y	N	Y	Y	Unknown	N	NA	Y	56
Ferrante et al., 2018 [39]	Y	Y	Y	Y	Y	N	Y	NA	Y	78
Ormel et al., 2018 [40]	Y	Y	Y	Y	Y	Unknown	Y	NA	Y	78
Golsteijn et al., 2018 [41]	Y	Y	N	Y	Y	N	Y	Y	Y	78
Mayer et al., 2017 [42]	Y	Y	N	Y	Y	N	N	Y	Y	67
Valle et al., 2017 [43]	Y	Y	N	Y	Y	Y	N	NA	Y	67
Short et al., 2017 [44]	Y	Y	N	Y	N	Unknown	Y	N	Y	56
Gnagnarella et al., 2016 [45]	Y	Y	Y	Y	N	Y	N	Y	Y	78

* [26,27] are two articles describing the same study.

## Data Availability

Data are available from the authors upon request.

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
