# Peer review of "mHealth Interventions to Promote a Healthy Diet and Physical Activity among Cancer Survivors: A Systematic Review of Randomized Controlled Trials"

_cancers, 2022, doi:10.3390/cancers14153816_

Round 1

Reviewer 1 Report

This was a systematic review of randomized controlled trials (RCTs) published through August 2021 that tested mHealth PA and/or dietary interventions among cancer survivors and report on feasibility, satisfaction, behavioral change, and/or quality of life (QOL) outcomes.

General comments:

1.     Don’t start a sentence with an integer.

2.     Punctuation comes before references.

Methods:

1.     Why did the search criteria begin in 2016? Was this when the first mHealth study was conducte?

2.     Define “cancer survivor” does this mean anyone with a diagnosis of cancer (pre-treatment, treatment-naïve, etc.)

3.     Change exclusion criteria from “cancer survivor” to “no diagnosis of cancer”

4.     Instead of “articles not focused on adult populations” change to “less than 18 years of age excluded” and justify why this age range excluded.

5.     I am confused by the way the following sentence reads: “22 studies, including 2 studies that each was reported by 2 articles”. Does this mean they published the same study in two different journals? Does it mean it was the same study but they reported on different outcomes? This is confusing to me.

6.     Did the article following PRISMA guidelines? This is typically a requirement for a systematic review (in addition to reviewing more than one database).

7.     Study quality assessment is typically conducted by at least two authors to prevent bias in reporting. How did you decide on these questions to assess study quality? Were studies excluded if they were not high quality?

8.     I would recommend adding a PICOS table for inclusion/exclusion.

Results:

1.       The language goes back and forth between “across the 22 unique studies” and “participants in 15 of the 20 studies”. Another example is on line 134 to 135. This is very confusing because I am unsure if there were 22 studies included in the systematic review or 20 studies included.

2.       In the methods there is a statement that “real time interventions were excluded” however, in the results there is a statement that “12 included in-person/telephone/video/e-mail coaching interventions” which sound like real-time interventions to me. Does this mean the two that were excluded were not mHealth interventions? If so, I recommend moving those two articles to the “not a mHealth category” of the exclusion criteria.

3.       In the results table 2, there are 2 diet and pa, 15 pa, 5 diet, and 9 quality of life interventions discussed. However, in the write up it states that two focused on diet only, 6 had diet and pa, and 14 pa only. I would recommend being consistent in these and how they are reported.

4.       For the studies that measured diet, how was it measured? Self-reported questionnaire? FFQ? 24-hour recalls? These are important components that the publication is lacking.

5.       Line 244-245 should be listed in the discussion. Not appropriate for the results section.

Author Response

Reviewer 1

Comments and Suggestions for Authors

This was a systematic review of randomized controlled trials (RCTs) published through August 2021 that tested mHealth PA and/or dietary interventions among cancer survivors and report on feasibility, satisfaction, behavioral change, and/or quality of life (QOL) outcomes.

General comments:

  1. Don’t start a sentence with an integer.

Response:  This has been changed.

  1. Punctuation comes before references.

Response: This has been changed.

Methods:

  1. Why did the search criteria begin in 2016? Was this when the first mHealth study was conducted?

Response:  We used 2016 as our cut-off date because our goal was to report the result on recent studies (in the past ~5 years), given how quickly mHealth technology has changed. In addition, most papers before 2016 included protocols with no results.

  1. Define “cancer survivor” does this mean anyone with a diagnosis of cancer (pre-treatment, treatment-naïve, etc.)

Response: Definition added. Page 2 line 77

  1. Change exclusion criteria from “cancer survivor” to “no diagnosis of cancer”

Response: Changed “cancer survivors” to “participants who were not diagnosed with cancer”. Page 2 line 96

  1. Instead of “articles not focused on adult populations” change to “less than 18 years of age excluded” and justify why this age range excluded.

Response: This has been changed. In this review, we wanted to focus on the adult population as the feasibility and acceptability of mHealth interventions may vary by age.

  1. I am confused by the way the following sentence reads: “22 studies, including 2 studies that each was reported by 2 articles”. Does this mean they published the same study in two different journals? Does it mean it was the same study but they reported on different outcomes? This is confusing to me.

Response:  For these two studies, each study was described in two paper that reported different outcomes. We have clarified this in the manuscript.

  1. Did the article following PRISMA guidelines? This is typically a requirement for a systematic review (in addition to reviewing more than one database).

Response: Yes. Description about PRIMA guidelines was added in the method section.

  1. Study quality assessment is typically conducted by at least two authors to prevent bias in reporting. How did you decide on these questions to assess study quality? Were studies excluded if they were not high quality?

Response: The questions for quality assessment are adapted from previously published scales.[1] We excluded studies that were not randomized controlled trials. Other than that criteria, we did not exclude studies based on their quality score. However, all included studies met at least five of the nine criteria.

  1. I would recommend adding a PICOS table for inclusion/exclusion.

Results: Can you please clarify what a PICOS table is? We are happy to add a table at the editors’ discretion if needed.

  1. The language goes back and forth between “across the 22 unique studies” and “participants in 15 of the 20 studies”. Another example is on line 134 to 135. This is very confusing because I am unsure if there were 22 studies included in the systematic review or 20 studies included.

Response: There are 22 studies included in the review, but not all the studies reported on every variable or outcome of interest. For example, only 11 studies reported the race/ethnicity of their participants. This is why the denominator changes and we have tried to clarify this throughout the text.

  1. In the methods there is a statement that “real time interventions were excluded” however, in the results there is a statement that “12 included in-person/telephone/video/e-mail coaching interventions” which sound like real-time interventions to me. Does this mean the two that were excluded were not mHealth interventions? If so, I recommend moving those two articles to the “not a mHealth category” of the exclusion criteria.

Response:  Studies were excluded if only real time interventions were used. If a study included both real time session (e.g. in-person/telephone/video/e-mail coaching) and other mHealth intervention tools (e.g. website, wearable trackers), the study was included in this review. We have italicized the word “only” in our methods for clarity.

  1. In the results table 2, there are 2 diet and pa, 15 pa, 5 diet, and 9 quality of life interventions discussed. However, in the write up it states that two focused on diet only, 6 had diet and pa, and 14 pa only. I would recommend being consistent in these and how they are reported.

Response: We apologize for the confusion. The studies in Table 2 are group by outcomes that were reported in each paper: change in a lifestyle score based on both physical activity and diet; change of physical activity; change of diet; and change of quality of life. Studies that examined physical activity and diet, but did not combine their outcome into one score, are listed under both the “physical activity” and “diet” section (for example, Chow et al. 2020). We have added a footnote to clarify this in Table 2.

  1. For the studies that measured diet, how was it measured? Self-reported questionnaire? FFQ? 24-hour recalls? These are important components that the publication is lacking.

Response: We have added this information. Among the five studies that reported change in diet, two studies measured diet using 24-hour diet record, one study measured diet using FFQ, one study measured diet using 4-day diet record, and one study measured diet using self-administered questionnaire (the Nutrition Questionnaire). There are also two studies that reported changed in a lifestyle score based on both physical activity and diet. These two studies both measured diet using validated FFQs.

  1. Line 244-245 should be listed in the discussion. Not appropriate for the results section.

Response: Unfortunately, we are not sure which sentence this is referring to. If needed, can you please clarify and we can make the adjustment?

Reviewer 2 Report

Overall, this is a useful and well-written review of mHealth interventions to promote improvements in diet and exercise behaviors of cancer survivors.  Suggestions for improvement are featured below:

Major concerns:

The abstract summarizes that mHealth interventions show acceptability and promise in promoting lifestyle changes among cancer survivors.  However, it is clear from the review that to date more success has been demonstrated with regard to physical activity than dietary change.  The need for more research as it relates to diet and weight management should be underscored in the abstract and also deserves a bit more embellishment in the discussion.  In addition, the need for research in populations that have yet to be “touched” by mHealth interventions, i.e., rural dwellers and older cancer survivors (particularly since the most recent statistics indicate that two-thirds of cancer survivors are age 65+ and older) also should be discussed.   

It appears that the RCT of an mHealth physical activity intervention among colon cancer survivors completed by Mayer et al. was overlooked (PMCID: PMC5911378).

Minor concerns:

The first sentence of the introduction should present updated numbers on number of cancer survivors (https://acsjournals.onlinelibrary.wiley.com/doi/full/10.3322/caac.21731)

Page 3: Line 106:  Suggest “assess” be replaced by “assessed.”

The authors might consider using left sided justification in their tables to avoid weird spacing.

Table 2 should be arranged by the type of intervention, not by the primary outcome and although QOL may have been listed as such, the studies reported should report on behavior change as well as QOL outcomes

Table 3 may be more easily interpreted if the results were arranged by the duration of the intervention and serve to reinforce the author’s point within the discussion that more research may be necessary in developing interventions that can support longer term behavior change.

Author Response

Reviewer 2

Comments and Suggestions for Authors

Overall, this is a useful and well-written review of mHealth interventions to promote improvements in diet and exercise behaviors of cancer survivors.  Suggestions for improvement are featured below:

Major concerns:

The abstract summarizes that mHealth interventions show acceptability and promise in promoting lifestyle changes among cancer survivors.  However, it is clear from the review that to date more success has been demonstrated with regard to physical activity than dietary change.  The need for more research as it relates to diet and weight management should be underscored in the abstract and also deserves a bit more embellishment in the discussion.  In addition, the need for research in populations that have yet to be “touched” by mHealth interventions, i.e., rural dwellers and older cancer survivors (particularly since the most recent statistics indicate that two-thirds of cancer survivors are age 65+ and older) also should be discussed.   

Response: We have added lines to the discussion about the relatively low number of studies with interventions focused on diet and lack of focus on older populations. Information about urban versus rural residence was not reported in the included studies.

It appears that the RCT of an mHealth physical activity intervention among colon cancer survivors completed by Mayer et al. was overlooked (PMCID: PMC5911378).

Response: Mayer et al. (PMCID: PMC5911378) has been added to the review.

Minor concerns:

The first sentence of the introduction should present updated numbers on number of cancer survivors (https://acsjournals.onlinelibrary.wiley.com/doi/full/10.3322/caac.21731)

Response: First sentence has been updated.

Page 3: Line 106:  Suggest “assess” be replaced by “assessed.”

Response: This has been done.

The authors might consider using left sided justification in their tables to avoid weird spacing.

Response: This has been done.

Table 2 should be arranged by the type of intervention, not by the primary outcome and although QOL may have been listed as such, the studies reported should report on behavior change as well as QOL outcomes

Response: Thank you for this suggestion. However, when we tried to make the table sorted by intervention (PA + Diet, PA only, Diet only) with columns for the different outcomes, the resulting table was very wide with many blank cells (since none of the PA only studies would report on diet, etc.). If the Editor recommends that Table 2 be sorted by the intervention, we can make this change.

Table 3 may be more easily interpreted if the results were arranged by the duration of the intervention and serve to reinforce the author’s point within the discussion that more research may be necessary in developing interventions that can support longer term behavior change.

Response: Table 3 has been rearranged by duration of the intervention (from the shortest duration to the longest duration).

Round 2

Reviewer 1 Report

I appreciate the opportunity to review the revised manuscript. In future reviews, it would be helpful if the author could list the page and line number of each edit make to the manuscript. This was challenging to review based upon a comment such as "this has been changed" which doesn't tell me HOW it was changed in the manuscript. 

  1. Why did the search criteria begin in 2016? Was this when the first mHealth study was conducted?

Response:  We used 2016 as our cut-off date because our goal was to report the result on recent studies (in the past ~5 years), given how quickly mHealth technology has changed. In addition, most papers before 2016 included protocols with no results.

Please add this information to the methods section. 

  1. Study quality assessment is typically conducted by at least two authors to prevent bias in reporting. How did you decide on these questions to assess study quality? Were studies excluded if they were not high quality?

Response: The questions for quality assessment are adapted from previously published scales.[1] We excluded studies that were not randomized controlled trials. Other than that criteria, we did not exclude studies based on their quality score. However, all included studies met at least five of the nine criteria.

Please add this as a limitation of the manuscript and add that studies were not excluded based on quality scores. 

  1. I would recommend adding a PICOS table for inclusion/exclusion.

Results: Can you please clarify what a PICOS table is? We are happy to add a table at the editors’ discretion if needed.

Information on PICOS table: https://libguides.murdoch.edu.au/systematic/defining#s-lib-ctab-22166366-4 

  1. In the methods there is a statement that “real time interventions were excluded” however, in the results there is a statement that “12 included in-person/telephone/video/e-mail coaching interventions” which sound like real-time interventions to me. Does this mean the two that were excluded were not mHealth interventions? If so, I recommend moving those two articles to the “not a mHealth category” of the exclusion criteria.

Response:  Studies were excluded if only real time interventions were used. If a study included both real time session (e.g. in-person/telephone/video/e-mail coaching) and other mHealth intervention tools (e.g. website, wearable trackers), the study was included in this review. We have italicized the word “only” in our methods for clarity.

Please add this to the discussion. Why were real-time only interventions included? 

  1. For the studies that measured diet, how was it measured? Self-reported questionnaire? FFQ? 24-hour recalls? These are important components that the publication is lacking.

Response: We have added this information. Among the five studies that reported change in diet, two studies measured diet using 24-hour diet record, one study measured diet using FFQ, one study measured diet using 4-day diet record, and one study measured diet using self-administered questionnaire (the Nutrition Questionnaire). There are also two studies that reported changed in a lifestyle score based on both physical activity and diet. These two studies both measured diet using validated FFQs.

Where was this added and what Nutrition Questionnaire was used? Please use a reference to the specific questionnaires (what FFQ, how was 24-hour diet record collected? by a trained researcher?). 

  1. Line 244-245 should be listed in the discussion. Not appropriate for the results section.

Response: Unfortunately, we are not sure which sentence this is referring to. If needed, can you please clarify and we can make the adjustment?

The sentence "these two studies by our team targeted both pa and diet change among prostate cancer survivors". Please remove "by our team" OR discuss this further in the discussion. 

Reviewer 2 Report

The authors have addressed my concerns

Author Response

Comments and Suggestions for Authors

The authors have addressed my concerns

Response: Thank you.